https://doi.org/10.1038/s42003-021-02266-z | **OPEN**
# Phasor-based hyperspectral snapshot microscopy allows fast imaging of live, three-dimensional tissues for biomedical applications

Per Niklas Hedde [1,2,3✉], Rachel Cinco[1], Leonel Malacrida [4,5], Andrés Kamaid[5] & Enrico Gratton [1,3✉]

Hyperspectral imaging is highly sought after in many fields including mineralogy and geology, environment and agriculture, astronomy and, importantly, biomedical imaging and biological fluorescence. We developed ultrafast phasor-based hyperspectral snapshot microscopy based on sine/cosine interference filters for biomedical imaging not feasible with conventional hyperspectral detection methods. Current approaches rely on slow spatial or spectral scanning limiting their application in living biological tissues, while faster snapshot methods such as image mapping spectrometry and multispectral interferometry are limited in spatial and/or spectral resolution, are computationally demanding, and imaging devices are very expensive to manufacture. Leveraging light sheet microscopy, phasor-based hyperspectral snapshot microscopy improved imaging speed 10–100 fold which, combined with minimal light exposure and high detection efficiency, enabled hyperspectral metabolic imaging of live, three-dimensional mouse tissues not feasible with other methods. As a fit-free method that does not require any a priori information often unavailable in complex and evolving biological systems, the rule of linear combinations of the phasor could spectrally resolve subtle differences between cell types in the developing zebrafish retina and spectrally separate and track multiple organelles in 3D cultured cells over time. The sine/cosine snapshot method is adaptable to any microscope or imaging device thus making hyperspectral imaging and fit-free analysis based on linear combinations broadly available to researchers and the public.

[1] Laboratory for Fluorescence Dynamics, University of California, Irvine, CA, USA. [2] Department of Pharmaceutical Sciences, University of California, Irvine, CA, USA. [3] Beckman Laser Institute & Medical Clinic, University of California, Irvine, CA, USA. [4] Departamento de Fisiopatología, Hospital de Clínicas, Facultad de Medicina, Universidad de la República, Montevideo, Uruguay. [5] Advanced Bioimaging Unit, Institut Pasteur of Montevideo and Universidad de la República, Montevideo, Uruguay. ✉email: phedde@uci.edu; egratton@uci.edu

Hyperspectral imaging is paramount to a vast array of fields including geosciences, environmental, and agricultural surveillance, quality control in industrial processes, astronomy, molecular biology, and biomedical imaging and diagnostics[1]. While the human eye perceives color as a mixture of three bands—red, green, and blue—hyperspectral imaging detects a much finer spectral structure, and can extend beyond the visible spectrum. Combined with microscopy and biological fluorescence[2], hyperspectral imaging has resolved biomolecular structures and functions in cells, tissues, and animal models. In humans, examples of medical applications include cancer detection with near infrared hyperspectral imaging[3], and diagnosis of retinopathy in the eye by detecting drops in oxygen consumption[4]. Current hyperspectral imaging methods typically split light with a dispersive element, such as a prism or grating before detection in a multichannel array at different spatial locations[5,6]. Because one spatial dimension is used to collect spectral information, spatial scanning must be used to collect a two-dimensional image[7]. Alternatively, light can be detected through a tunable filter at different times (spectral scanning) to yield an emission spectrum in each pixel/voxel[8,9]. While working well, such scanning approaches are generally time consuming and reduce light throughput. Snapshot methods have been developed to overcome these limitations, for example, using image mapping[10] or Fourier transform-based spectral decoding[11,12]. Despite being much faster, these methods are limited in spatial and/or spectral resolution, are computationally demanding, and devices are very expensive to manufacture.

The phasor approach to hyperspectral imaging maps a complicated spectrum on a 2D plot (phasor plot) with a pair of Fourier sine and cosine transforms. Each image pixel/voxel can be represented on this phasor plot with the angular position (phase) determined by the center of mass of the emission, and the distance from the center (modulation) determined by the width of the spectrum. Due to the law of linear addition, shifts in the emissions between multiple components can be easily quantified without knowledge of the underlying spectra. For example, the signals from two components mixed at different ratios will all localize to a line on the phasor plot. Even without knowledge of the positions of the end points (i.e., the pure components), their relative ratio can still be calculated from the relative position along the connecting line. Similarly, three components will form a triangle, and four components will form a quadrilateral geometry on the phasor plot. Again, their relative ratio can be calculated from the relative position. Another advantage of the phasor transformation is the reciprocity principle in which a region of interest can be selected in both the image and the phasor plot, to identify the corresponding phasor components and regions of interest. These properties are very valuable considering complex emission within a pixel and does not require traditional unmixing.

Here, phasor-based hyperspectral snapshot microscopy performs the phasor transformation directly in hardware by passing the light though a pair of optical filters with transmission spectra in the form of a sine/cosine period[13] in the desired wavelength detection range. With the only requirement being the replacement of standard-sized emission filters, our approach can be implemented on any commercial or home-built imaging system including all camera-based and laser scanning microscopes. Our approach does not need a specialized spectral detector and, importantly, does not use computational unmixing. Instead, fit-free phasor-based analysis[14,15] based on linear combinations was used to quantify biological samples. In complex biological environments, the spectral components often depend on the environment presenting a challenge for traditional spectral unmixing[16,17] that demand a priori information. If desired, the individual spectral components can still be unmixed with conventional algorithms[18,19].

With a light sheet microscope[20,21], we show how phasor-based hyperspectral snapshot microscopy can achieve a 10–100-fold improvement in imaging speed (1.3–3 images/s) over conventional hyperspectral laser scanning microscopy (<0.2 images/s)[5,9,15], line-scanning light sheet microscopy (0.14–0.67 images/s)[7], and light sheet microscopy with tunable filters (0.1 images/s)[8]. Additionally, our approach preserves the full camera field of view compared to image mapping snapshot methods (e.g., field of view divided into $5 \times 6$ sub-images)[10], and maintains high light collection efficiency (64% average sine/cosine filter transmission). For biological imaging, ultrafast 5D ($x$, $y$, $z$, $\lambda$, $t$) imaging with minimal photobleaching enabled tracking of organelles in 3D cultured cells as shown in this manuscript. In another biological application, ultrafast phasor-based hyperspectral imaging and linear combination analysis also allowed metabolic imaging of live, three-dimensional mouse tissues, and the quantification of water dipolar relaxation in the zebrafish eye not achievable with conventional solutions, that require knowledge of the spectra of the underlying components. We further demonstrate how fit-free phasor analysis based on linear combinations was able to resolve subtle differences between cell types in a complex tissue, the developing zebrafish retina. While the spatiotemporal resolution of light sheet microscopy can take full advantage of the sine/cosine filter method, our snapshot method is adaptable to any microscope or imaging device. This method can make snapshot hyperspectral imaging and phasor-based analysis broadly available to researchers and the public.

## Results

**Concept and advantages of phasor-based hyperspectral snapshot microscopy for biomedical research.** In conventional hyperspectral imaging, emission light is optically split into a large number of spectral channels, that are either spatially[5,6] (Fig. 1a) or spectrally scanned[8,9] (Fig. 1b). The detector can be a linear array for point scanning methods (e.g., laser scanning microscopy)[6] or a two-dimensional array such as a camera[7]. As only a maximum of two ($x$, $y$, $x$, $\lambda$, or $y$, $\lambda$) of the three dimensions required to form a full hyperspectral image ($x$, $y$, $\lambda$) can be acquired simultaneously, these methods are inherently slow. Snapshot methods (Fig. 1c) such as image mapping[10] or spectral decoding[11] can acquire hyperspectral images without scanning, but image remapping/decoding compromises spatial and/or spectral resolution, is computationally demanding, and devices are often associated with high manufacturing cost. In contrast, our approach uses a pair of filters with sine/cosine transmission spectra in the desired detection range (here 400–700 nm) for direct, in-hardware transformation of the spectral information to frequency space (phasor space) (Supplementary Fig. S1). Hence, no changes to the optics, the detector, and the image acquisition procedure are needed and neither is any computational decoding of the signal (Fig. 1d, e). Therefore, full advantage can be taken of the high imaging speeds provided by state-of-the-art imaging approaches. With light sheet microscopy[20,21], three-dimensional hyperspectral or 4D imaging ($x$, $y$, $z$, $\lambda$) can reach the same high speeds (>1 frame/s) as bandpass filter imaging, while the highlight throughput of the sine/cosine filters (64%) also ensures minimal photobleaching and phototoxicity, additional benefits of light sheet imaging (Fig. 1f).

To demonstrate and quantify the spectral resolution and accuracy of the sine/cosine filters by comparison with traditional methods, we prepared fluorescent dye solutions for spectral fingerprinting using a commercial laser scanning microscope with a 32 channel spectral detector as well as the sine/cosine filter

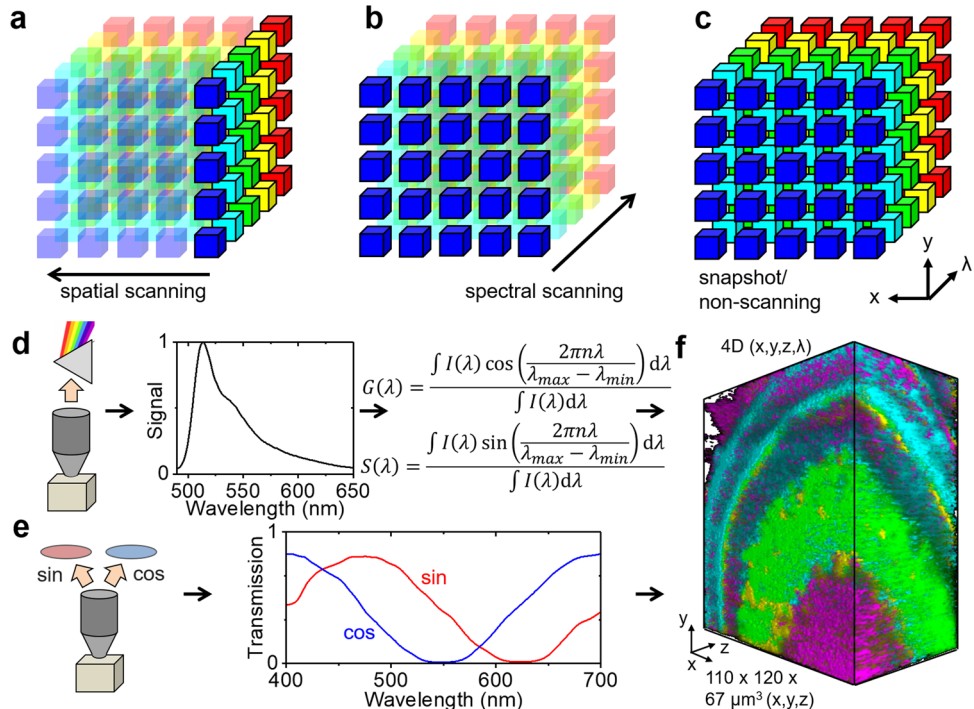

**Fig. 1 Concept and advantages of phasor-based hyperspectral snapshot microscopy. a, b** Conventional hyperspectral imaging methods rely on slow spatial or spectral scanning. **c** Faster snapshot methods exist but often compromise spatial and/or spectral resolution and can be computationally demanding. **d** In conventional phasor-based hyperspectral imaging, an emission spectrum is acquired in each pixel/voxel that is then computationally transformed to the spectral phasor plot for analysis by a sine/cosine transformation. **e** Instead, the sine/cosine filter approach to hyperspectral imaging directly transforms the fluorescence emission to spectral phasor coordinates in hardware, no spectral detector and image data post-processing are required. **f** The resulting substantial improvement in image acquisition and data processing speed allows the generation of 4D data ($x$, $y$, $z$, $\lambda$) in seconds taking full advantage of the high light throughput and minimal photobleaching of light sheet microscopy.

method. Figure 2a shows the positions of six dye solutions (NADH, FAD, Rhodamine 110, Rhodamine 6G, 5-TAMRA, and Alexa 594) on the spectral phasor plot as measured by the sine/cosine filter method. In the phasor plot, the polar coordinates phase (angular position) and modulation (radial position) represent the center emission wavelength and width of the spectral signal. We then compared the sine/cosine filter method with a commercial laser scanning microscope with a 32-channel spectral detector (Fig. 2b), and found no differences when graphing both parameters (phase and modulation) obtained by either method against each other (Fig. 2c). A table summarizing all parameters can be found in Supplementary Fig. S2. Supplementary Fig. S3 shows the raw image data and additional phasor plot locations of a total of eleven dyes. Additionally, we tested one of the major strengths of the phasor approach, the rule of linear additions, which allows quantifying the ratio of multiple (spectral) components present in the same spatial location without a priori information[14]. Mixing two dyes (Rhodamine 110 and Atto 590) at different proportions, all fractions were found along a connecting line between the pure species as expected (Fig. 2d). We note that, as with all spectral detectors, the detection range (here 400–700 nm) is truncated; missing a spectra's long wavelength tail will slightly shift the center of mass of the emission as determined by phasor analysis and increase the perceived modulation. Yet, within a limit, this shift will not affect the rule of linear addition, as shown in Supplementary Fig. S4.

**Hyperspectral phasor-based cellular fingerprinting in complex biological tissue.** In order to demonstrate the advantages of our method in a complex biological sample, we applied ultrafast phasor-based hyperspectral snapshot microscopy in zebrafish

embryos, a prime model to study developmental processes that is also widely used to model human diseases. We took advantage of the Spectrum of Fates approach (SoFa), in which a transgenic line was generated to study zebrafish retinal development by driving the expression of differently tuned fluorescent proteins (FPs) from the promoters of a set of transcription factors[22]. In SoFa embryos, each FP is expressed in one or multiple cell types resulting in a different fluorophore ratio associated with each of the major cell types. This difference in FP ratios creates a unique spectral fingerprint for each cell type. We imaged 72 h of post-fertilization (hpf) embryos of SoFa1 zebrafish[22] expressing Crx:gapCFP, Ptf1a:cytGFP, and Atoh7:gapRFP, together with SYTOX Orange (SO) nucleic acid stain added after embryo fixation. Importantly, the emission spectra of CFP/EGFP and RFP/SO completely overlap and could not be separated by conventional bandpass filters (Supplementary Fig. S5). Thus we applied 4D ($x$, $y$, $z$, $\lambda$) light sheet microscopy with the sine/cosine filter method (Fig. 3a–j). Noteworthy, the entire 4D data set (100 $z$ sections) was taken in ~1 min as opposed to ~1 h on our laser scanning system. The spectral phasor shows the linear combinations of the four fluorophore emissions with the theoretical positions of the pure species marked by colored circles (Fig. 3b).

Again, phasor analysis using linear combinations is fit-free/deconvolution-free and does not require any a priori assumptions on the components, very important in unfixed or developing biological samples, where the environment can modify the emission spectra of the fluorophores as a function of time. In Fig. 3b, the phasor histogram was divided into two halves with the division running though the midpoints between GFP/CFP and SYTOX/RFP. The voxels in each half were then color coded according to their position along the connecting lines of the two fluorophore pairs CFP/RFP and GFP/SO using the indicated

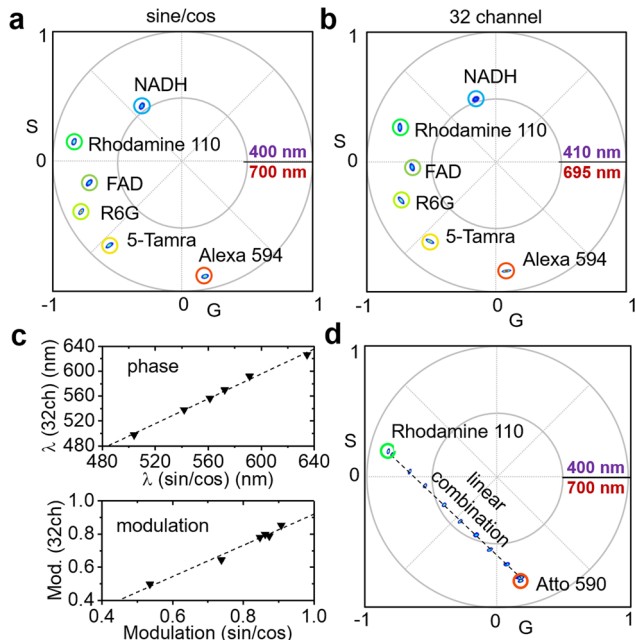

**Fig. 2 Validation of phasor-based hyperspectral snapshot microscopy.**
**a** Spectral phasor plot of six different dye solutions measured with sine/cosine filters (NADH, FAD, Rhodamine 110, Rhodamine 6G, 5-TAMRA, and Alexa 594). **b** Spectral phasor plot of the same six dye solutions measured with a commercial 32 channel hyperspectral detector. **c** Phases ($\lambda$) and modulations obtained by the sine/cosine method graphed against the values measured by conventional 32 channel hyperspectral detection with computational phasor transformation; perfect agreement was observed. **d** Spectral phasor plot of two dyes (Rhodamine 110 and Atto 590) mixed at different proportions. All fractions were found along a connecting line between the pure species as expected by the law of linear combinations.

color maps (cyan/magenta, green/yellow). This color code was applied to the respective 4D maximum intensity projection (MIP) (Fig. 3c); a 360° rotation of a color coded 4D MIP is shown in Supplementary Movie M1. Individual colors of zoomed-in 3D MIPs (white box in Fig. 3c) are shown in Fig. 3d–g, the 4D composite MIP is shown in Fig. 3h. The intensity profiles for each color (Fig. 3i) matched the expected cellular composition (Fig. 3j)[22].

For detailed cellular fingerprinting based on spectral information, the reciprocity of the phasor transformation allowed us to select regions in the image and build the corresponding phasor histograms. Using the intensity image, we selected layers corresponding to the different cell types and layer organization of the zebrafish retina at this embryonic stage, see Fig. 4a, b (the scheme is a reprint of Fig. 3j to improve readability of Fig. 4; M1/PR, photoreceptors; M2/HC, horizontal cells; M3/BC, bipolar cells; M4/AC, amacrine cells; dAC, displaced amacrine cells; and M5/RGC, retinal ganglion cells). The expression of the reporters is complex, with several cell types co-expressing more than one FP, no simple "single color" pattern was expected for most cell types. For example, PR can express CFP and RFP, and some HC can express GFP and RFP at lower levels as the level may vary depending on the cell cycle phase or differentiation state. However, as the phasor plot positions of the pure components (CFP, GFP, RFP, and SO) are defined by their spectra, the resulting mixtures must localize within the perimeter of the quadrilateral defined by the pure species. For each image pixel, the relative fractions contributing to the detected signal can be determined based on the distance to the pure components. This geometrical linear combination approach based on the phasor

plot distributions is further discussed in Supplementary Fig. S6, and the underlying principle of phasor analysis is discussed and step-by-step instructions can be found in Ranjit et al.[14]. In the M1/PR region (Fig. 4c, d), the spectral phasor plot identified the linear combination between CFP and SO (labeled CFP-SO) and a group of voxels with a fraction of CFP/SO plus RFP (labeled CFP-RFP), which corresponded to the expected pattern of high expression of CYP and low expression of RFP in the PR. In the M2/HC region (Fig. 4e, f), two linear combinations were found, CFP-SO and GFP-SO, which was expected to show very strong CFP from PR cells and GFP from HC (and sometimes low expression of RFP, which in this particular data set was not observed). The M3/BC phasor plot (Fig. 4g, h) showed a dense voxel cluster at the center between all components, associated with a linear combination of CFP and SO (labeled CFP-SO) highlighted with the cyan polygon. Two more components were detected and highlighted with the green and magenta polygons. The cyan voxels corresponded to the BC layer with strong CFP expression, the voxels in the green polygon identified amacrine cells of the adjacent HC layer. The few voxels in the magenta polygon likely corresponded to the described BC cells with CYP/SO and low expression of RFP, visible in some areas. This result reflects high levels of CFP expression versus low levels of RFP in this layer. The M4/AC phasor plot (Fig. 4i, j) identified a linear combination of GFP and SO. These voxels were highlighted by the black polygon labeled as GFP-SO and color coded with a green-yellow gradient in the corresponding image. In addition, there was a small group of voxels towards the CFP location, identified with the cyan polygon. The GFP-SO gradient corresponded well to the expected expression of GFP in amacrine cells together with the nuclear SO stain. The cyan region highlighted a group of voxels associated with axons of BC cells expressing CFP. The M5/RGC phasor plot (Fig. 4k, l) identified a main cluster of voxels associated with a linear combination of RFP/SO, that was highlighted by the magenta polygon. Two more groups were observed with either GFP or CFP and different fractions of SO (green and cyan polygons). The magenta polygon selection corresponded to RGC, while the green polygon selection was consistent with displaced amacrine cells (dAC), and the cyan polygon selection with BC neurites (cyan arrowhead).

We note that by using two-photon excitation at 920 nm in the SPIM microscope instead of multiple lines of visible light, no notch filters and/or dichroic mirrors were needed that would have reduced light throughput and distorted the spectral phasor linear combinations by adding gaps into the 400–700 nm detection range. Exemplary raw camera images of the SoFa fish retina are shown in Supplementary Fig. S7.

**5D imaging enables spectral separation and tracking of organelles in live cells.** One of the most desired applications of hyperspectral imaging in biological samples is to be able to identify different structures of interest, such as cell organelles or proteins, and follow their dynamics in live tissues[23]. Conventional approaches include the use of several bandpass filters specific to the emissions of different fluorophores. While this approach works well with few probes/channels, it requires more specialized filter sets, narrow-band probes, and wider excitation/detection ranges to spectrally space many probes. Often, probes with the desired spectral characteristics are not available or overlap with intrinsically fluorescent molecules such as the cell metabolites nicotinamide adenine dinucleotide (NAD) in its reduced form NADH and flavin adenine dinucleotide (FAD). The addition of many channels can also slow image acquisition. Hyperspectral imaging can overcome these limitations by spectrally separating components with strongly overlapping emissions.

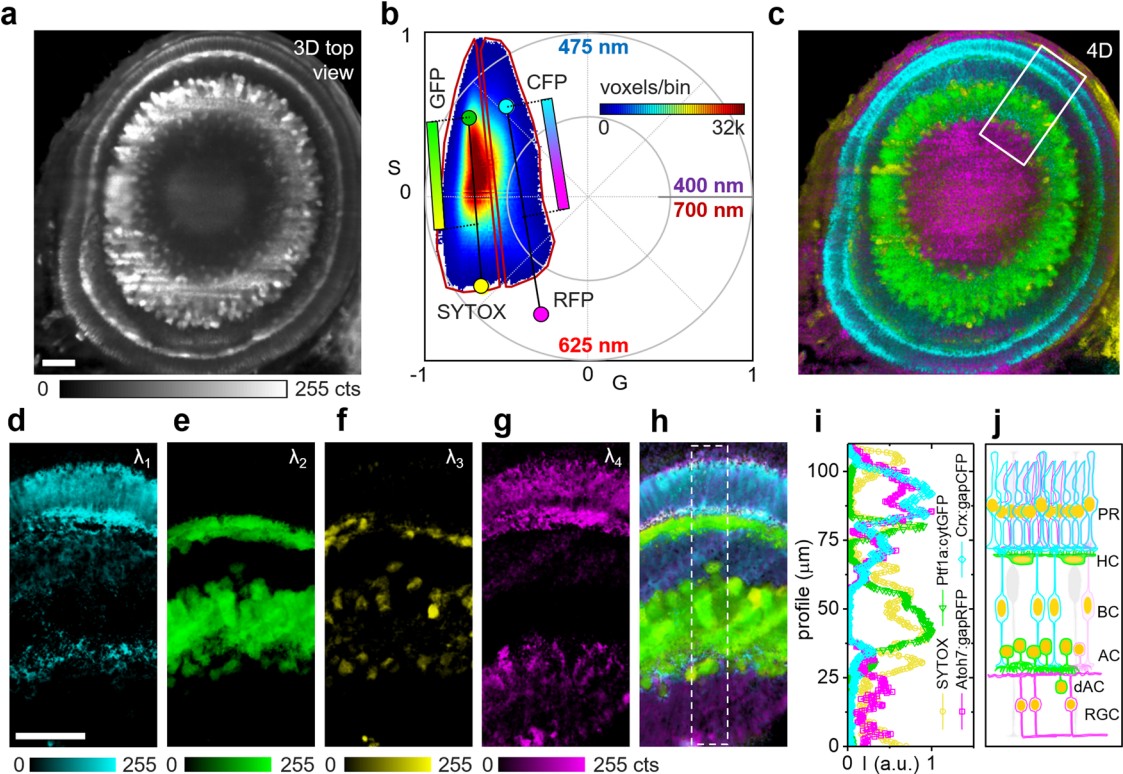

**Fig. 3 Retinal hyperspectral images of a transgenic SoFa 72 hpf zebrafish. a** Sliced maximum intensity projection (MIP) of 3D fluorescence image data showing the eye of a fixed 72 hpf SoFa zebrafish expressing Crx:gapCFP, Ptf1a:cytGFP, Atoh7:gapRFP plus SYTOX Orange nuclear stain. **b** Phasor plot of the total 151 z sections acquired by fluorescence collection though sine/cosine filters, color-coded circles indicate the theoretical locations of the pure fluorophore emissions. The phasor histogram was divided into two halves, the voxels in each half were color coded according to their position along the connecting lines of the two fluorophore pairs (CFP/RFP, GFP/SYTOX) with the indicated color maps (cyan/magenta, green/yellow). **c** MIP of 4D image data of the view shown in **a** color coded according to the voxel positions in **b** to indicate the dominant component. **d**–**h** Zoomed-in MIPs of 20 slice 3D fluorescence volume renders of the region indicated in **c** spectrally separated according to the voxel positions in **b**. **i** Normalized intensity profiles within the region marked in **h** indicative of the retinal cellular composition. **j** Cartoon of the underlying cell composition (PR photoreceptors, HC horizontal cells, BC bipolar cells, AC amacrine cells, dAC displaced amacrine cells, RGC retinal ganglion cells). All zebrafish labels were excited simultaneously via two photon absorption at 920 nm. Representative fish from three imaging sessions on three different days. Scale bars, 25 μm.

We therefore show spectral separation and organelle tracking by the sine/cosine filter approach in a biological sample. NIH 3T3 cells were seeded on a layer of collagen followed by labeling of the nucleus, Golgi, mitochondria, and lysosomes with live cell stains (Fig. 5a). The organelle stains were selected to have a strong spectral overlap such that conventional three-dimensional imaging with bandpass filters (3.5D–3D in few spectral channels) would struggle to distinguish Golgi, mitochondria, and lysosome staining (Fig. 5b–e). Instead, phasor-based hyperspectral snapshot microscopy could clearly distinguish the different organelles (Fig. 5f–l). For spectral separation in four windows, the phasor plot emission wavelengths were mapped in four wavelength ranges as indicated in Fig. 5f. The total 3D fluorescence intensity MIP is shown in Fig. 5g, individual unmixed components are shown in Fig. 5h–l, shows the 4D overlay. A 360° rotation of the color mapped 4D MIP is shown in Supplementary Movie M2. Due to the high temporal resolution of the sine/cosine filter approach, it was possible to track individual lysosomes by 5D (x, y, z, λ, t) sine/cosine light sheet imaging, see Fig. 5m–q and Supplementary Movie M3. For this purpose, 4D hyperstacks were acquired in 16 s at 2 min intervals. Another example set of images including the raw sine/cosine data is shown in Supplementary Fig. S8. To help define the spectral window ranges, the emissions of single stained cells were characterized in Supplementary Fig. S9. Using 2-photon excitation at 870-nm, all fluorophores could be excited simultaneously with a single wavelength. This approach avoided the need for additional dichroic mirrors and notch filters in the light path that would have reduced the light throughput, and degraded the spectral sensitivity by blocking certain wavelength ranges. While bandpass filtering might have been improved with the addition a narrower filter specific to BODIPY FL emission, the 17-nm emission peak shift of LysoTracker Red with respect to MitoTracker Orange was insufficient for spectral separation with a fourth filter.

**Metabolic imaging of live mouse tissues.** Metabolic imaging of live samples is an emerging field with very promising applications in biomedicine. Known as the Warburg effect[24], cancer cells alter their metabolism to promote growth, proliferation, and survival by increased glucose uptake and fermentation of glucose to lactate. While the exact function of the Warburg effect remains unclear, it has been found that stem cells preferentially undergo glycolysis whereas differentiated tissues typically consist of more oxidative phosphorylation-active cells[25], leading to the cell reversal hypothesis that cancer is a (partial) reversal of a differentiated cell to a non-differentiated stem-like state. Fluorescence of the cell metabolites NADH and FAD has been shown to discriminate between glycolysis and oxidative phosphorylation[26,27], which enables cancer detection by hyperspectral imaging. Here, we demonstrate ultrafast phasor-based hyperspectral snapshot microscopy for metabolic imaging by quantifying metabolism in ex vivo mouse intestinal crypts (Fig. 6a–h).

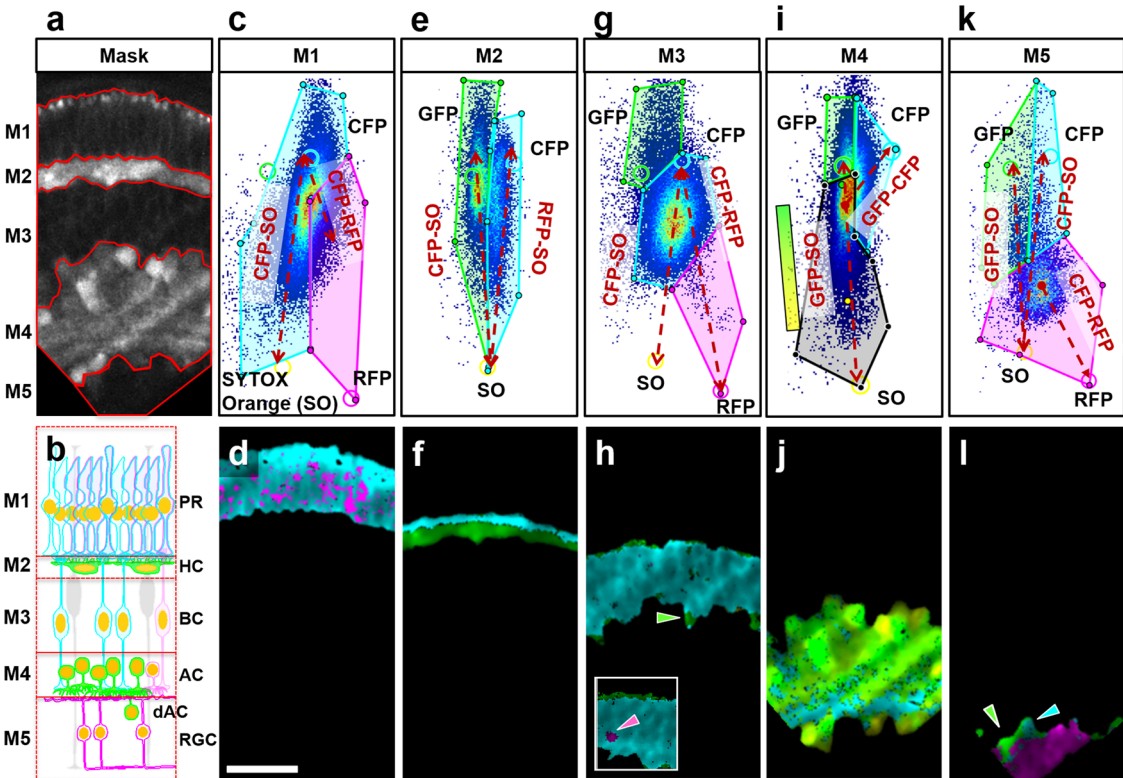

**Fig. 4 Hyperspectral cellular fingerprinting reveals zebrafish retinal composition. a** Fluorescence intensity image showing the region of the retina used for segmentation phasor analysis; red lines indicate masks (M1–M5) for the cell layers of the SoFa fish retina. **b** Scheme of the different cell types and layer organization of the zebrafish retina at this embryonic stage. **c** M1 phasor plot identifying the linear combination between CFP and SYTOX Orange (SO) (labeled CFP-SO) and a group of voxels with a fraction of CFP/SO plus RFP (labeled CFP-RFP). **d** Pseudocolored image representing the voxels selected inside the polygons. The pattern corresponded well with the expected high expression of CFP and low expression of RFP in the PR. **e** M2 phasor plot identifying two linear combinations, CFP-SO and GFP-SO. Polygon voxel selections highlight (**f**) CFP expressed in the upper boundary of the HC layer in addition to strong GFP expression from HC. **g** M3 phasor plot showing a big voxel cluster at the center between all components, associated with a linear combination of CFP and SO (labeled CFP-SO) marked with the cyan polygon. **h** The few voxels selected by the green polygon identified an amacrine cell (green arrowhead) of the adjacent layer (HC), showing the high spectral sensitivity of the phasor plot in each voxel. Additional weak expression of GFP and RFP was detected in this layer as well (green and magenta polygons), visible in some areas (inset, magenta arrowhead). **i** M4 phasor plot identifying the linear combination of GFP and SO highlighted with a black polygon and colored with a gradient color scheme (green-yellow) labeled as GFP-SO. This layer included a small group of voxels towards the CFP location, identified with the cyan polygon. **j** The pseudocolored image showed that the GFP-SO gradient corresponded well to the expected expression of GFP in amacrine cells, together with the nuclear SO stain. The cyan region highlighted a group of voxels at the bottom of the pseudocolored image, associated with axons of BP cells expressing CFP. **k** M5 phasor plot identifying a main cluster of voxels associated with a linear combination of RFP/SO highlighted by the magenta polygon. Two more groups were observed in this layer with either GFP or CFP and different fractions of SO (green and cyan polygons). **l** The pseudocolor image revealed that the magenta polygon selection corresponded to RGC, while the green polygon selection was consistent with displaced amacrine cells (dAC) (green arrowhead), and the cyan polygon selection with BP neurites (cyan arrowhead). Scale bar, 25 μm.

In the intestine, a stem cell population located in crypts allows for replacement of tissue digested in the lumen (Fig. 6g). The stem cell population preferentially undergoes glycolysis whereas the differentiated tissue consists of more oxidative phosphorylation-active cells[26]. This difference in metabolism is indicated by the relative amounts of NADH and FAD that can be quantified by the optical redox ratio, $rr$, defined as the ratio of NADH to FAD fluorescence[28]. Figure 6a, d show the 3D fluorescence intensity image of individual mouse intestinal crypts, the corresponding phasor plots are shown in Fig. 6b, e. While NADH is predominantly fluorescent in the range of 420–500 nm, the emission of FAD resides in the 520–600 nm range. From the phasor coordinates, we calculated the optical redox ratio as the projection along the trajectory shown in the phasor plot with the endpoints indicated by solution measurements of NADH and FAD (Supplementary Fig. S10). We note that the phasor position in the live mouse tissue is shifted towards the center of the phasor plot indicating a broader emission spectrum as pure NADH and

FAD in solution. This shift is a result of the much more complex live tissue environment with a distribution of free and bound forms of both NADH and FAD that broaden the emission[29]. Importantly, while imaging with two bandpass filters chosen based on pure (free) NADH and FAD may cut off these components, phasor analysis in the full emission range can robustly measure the NADH/FAD trajectory and without demanding a priori spectral information. The color coded 4D MIPs are visualized in Fig. 6c, f. In both examples, a clear gradient in the redox ratio was observable in axial direction. To quantify, we graphed the redox ratio of Fig. 6c along the $x$, $y$, and $z$ directions in Fig. 6h, each data point represents the average of a single 2D slice. In lateral direction ($x$, $y$), the redox ratio remained mostly uniform, while the gradient was confirmed in axial direction ($z$). This transition in the redox ratio recapitulates the metabolic change from the bottom (stem cell population) to the top part (differentiated cell population) of the intestinal crypt[26]. Such 4D visualization and quantification would be near-

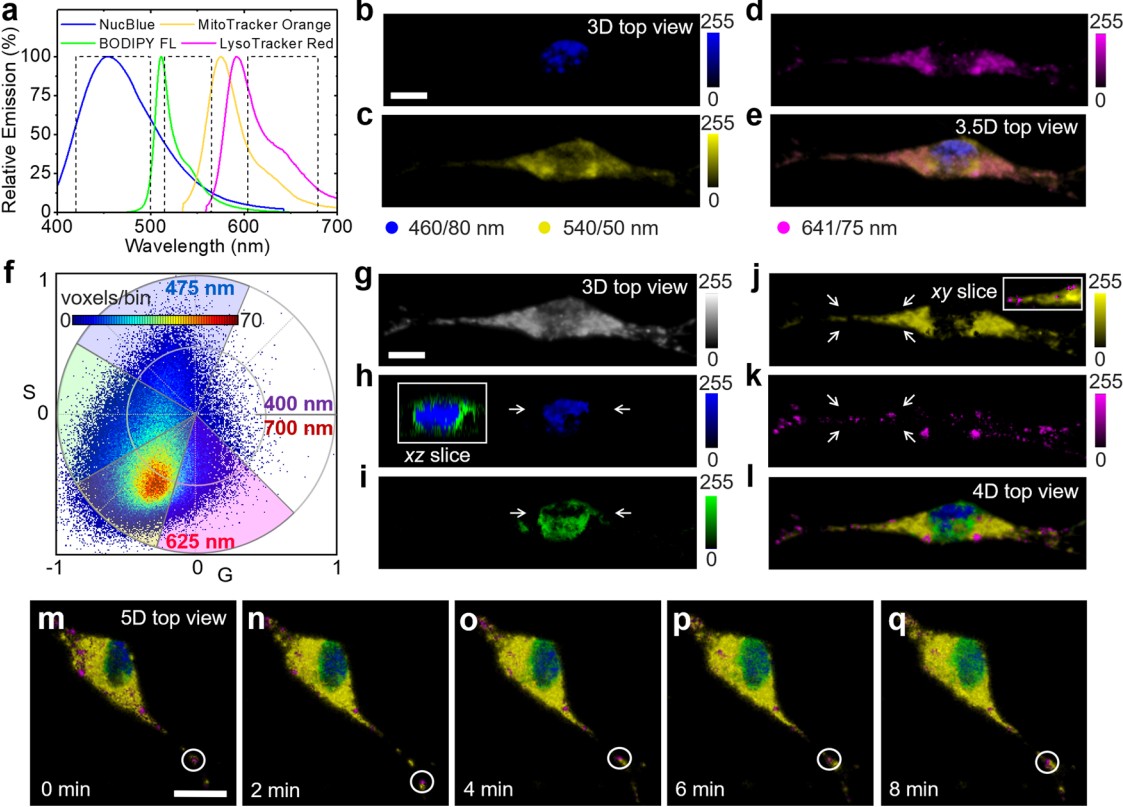

**Fig. 5 5D imaging enables spectral separation and tracking of organelles in live cells. a** The cell nucleus, Golgi, mitochondria, and lysosomes of NIH 3T3 cells were labeled with NucBlue, BODIPY FL, MitoTracker Orange, and LysoTracker Red (dye spectra raw data, Invitrogen) before ultrafast hyperspectral light sheet imaging with sine/cosine filters. **b–e** MIPs of 3D data and their overlay obtained by imaging in three spectral windows with bandpass filters (3.5D: 3D in channels DAPI, FITC, TexasRed, dashed lines in **a**; this conventional approach was unable to discern between Goligi, mitochondria, and lysosomes. **f** Spectral phasor plot obtained by fluorescence collection though sine/cosine filters of the same cell. **g** Total 3D MIP. **h–k** Unmixed MIPs color coded with respect to their emission center of mass wavelengths as indicated in the phasor plot (**f**). **l** 4D overlay, spectral phasor analysis could clearly distinguish all four organelle types. Insets in **h**, **j** show single, color-merged *xz/xy* planes to illustrate the position of nucleus/Golgi and mitochondria/lysosomes with respect to each other. **m–q** 5D (*x, y, z, λ, t*) hyperspectral time lapse sequence enabling lysosome tracking (white circles). All cell stains were excited simultaneously via two photon absorption at 870 nm. Representative cells from three imaging sessions on three different days. Scale bars, 10 μm.

impossible with conventional hyperspectral scanning techniques. During the hour-long sample-scanning process, the metabolism of ex vivo tissue would have deteriorated with gradients skewed by the onset of tissue necrosis. With 4D hyperstack imaging in under 1 min per sample, ultrafast phasor-based hyperspectral snapshot microscopy provides access to a spatiotemporal scale suitable for metabolic imaging.

**Cell-wide water dipolar relaxation quantified in live zebrafish embryos.** In addition to spectral unmixing, tracking, and metabolic imaging, hyperspectral microscopy can also be used to study complex photophysics and enable their application to biomedical studies. Here, we quantified cell-wide water dipolar relaxation in the developing live zebrafish embryo with the solvatochromic probe ACDAN (6-acetyl-2-dimethylaminonaphthalene). In addition to exhibiting different affinities for hydrophobic/hydrophilic environments[30], ACDAN fluorescence is sensitive to water dipolar relaxation phenomena occurring during their nanosecond fluorescence lifetime causing a shift of the pre-dominantly blue emission in an unrelaxed environment towards more green emission in a relaxed environment such as pure water (Supplementary Fig. S10)[31,32]. Using the rule of linear combination of the phasor, the emission of ACDAN can be spectrally and spatially resolved to quantify cell-wide water dipolar relaxation.

A zebrafish embryo was incubated with 50 μM ACDAN for 12 h and subjected to hyperspectral light sheet imaging 72 h of post-fertilization (hpf) by two-photon excitation with 800-nm light. The 3D fluorescence intensity MIP of the zebrafish eye is visualized in Fig. 7a, the spectral phasor plot for all voxels (4D—*x, y, z, λ*) is shown in Fig. 7b. From the phasor plot coordinates, we calculated the dipolar relaxation index, dr, defined as the ratio of the green to blue emission projected along the trajectory shown in the phasor plot with the endpoints indicating ACDAN in the relaxed and unrelaxed state. Unlike conventional unmixing algorithms, using linear combinations the phasor analysis does not require a priori information on each component nor a priori knowledge of the total number of components. Hence, the spectral phasor can overcome the limitations of general polarization (GP) analysis that quantifies the signal ratio detected in two discrete color channels[33]. The assumption of only two components with fixed spectra is a problem when working with probes such as ACDAN, where the emission maximum and shape depend on the system under study. For example, ACDAN emission reported in yeast[32] is more blue-shifted compared to our experiments in the zebrafish. Importantly, the phasor position in the zebrafish tissue was slightly shifted towards the center of the phasor plot indicating a wider emission spectrum than pure ACDAN in solution, a result of other tissue components (e.g., flavins) adding a spectrally broad autofluorescent background.

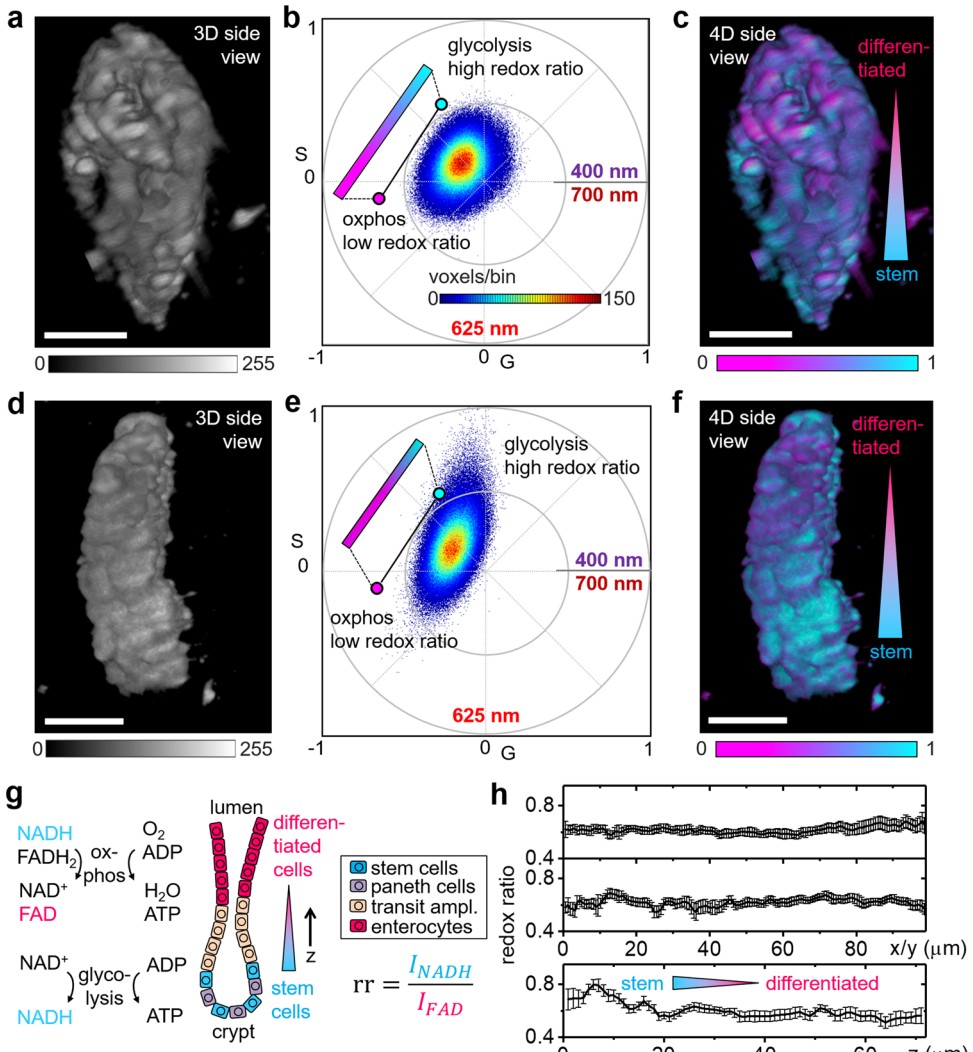

**Fig. 6 Metabolic imaging of live mouse tissues. a–f** The colon was dissected from a C57BL/6N mouse and embedded in Cultrex Matrix for sine/cosine hyperspectral imaging of NADH and FAD fluorescence. **a, d** MIPs of the 3D intensity data obtained by two photon excitation of NADH and FAD fluorescence with 740-nm light. **b, e** Spectral phasor plots obtained by collecting though sine/cosine filters. **c, f** MIPs of the 4D hyperspectral data color coded according to the voxel position along the phasor plot trajectory of NADH/FAD. **g** Molecular pathways for oxidative phosphorylation and glycolysis and schematic of the cellular composition in the intestine and preferred metabolic pathways; the optical redox ratio, $rr$, is defined as the ratio of NADH and FAD fluorescence intensities. **h** Graphs of the optical redox ratios, $rr$, along the $x$, $y$, and $z$ axes of the data visualized in **c**. Each point represents the average in one $yz$, $xz$, and $xy$ plane, error bars indicate standard deviations of the voxel redox ratios in each slice. A gradient in metabolic activity was found in axial direction. Representative images of five different crypts from two different animals imaged on two different days. Scale bars, 50 µm.

These components are included in the phasor analysis based on linear addition of all components. The color coded 4D MIP is visualized in Fig. 7c, a single $z$ section is shown in Fig. 7d. In Fig. 7e, the heterogeneity in the eye due to changes in cellular water dipolar relaxation was quantified across the eye architecture[34]. Clear differences were observed in the developing zebrafish eye with spectrally very blue ACDAN fluorescence in the aqueous humor, which corresponds to a lower $dr$ indicating low solvent (water) relaxation.

## Discussion

In summary, here we demonstrate ultrafast phasor-based hyperspectral snapshot microscopy with a light sheet microscope applied to several important biological questions. The method is capable of resolving spectral changes in vivo and ex vivo to reveal and quantify structures and mechanisms in complex biological systems with a spatiotemporal resolution not accessible by conventional approaches. With hardware changes

limited to installing a pair of emission filters, sine/cosine phasor-based hyperspectral imaging can easily be implemented on any microscope without extensive hardware modifications and without compromising the number of available image pixels/voxels as with other snapshot methods. The light throughput of the sine/cosine method (64%) is on par with much slower grating-based spectrographs (<70%) and far superior to tunable filters where only a fraction (typically < 10%) of the total emission is acquired at a time. Moreover, the sine/cosine filter solution is of marginal cost (~$2k) in comparison with commercial hyperspectral cameras (>$50k) as well as home built systems with custom optics[7,10,12]. Imaging spectrographs are of moderate cost (>$8k), but are slow and integration and synchronization with other system hardware can be tedious. While tunable filters (>$2k) are relatively easy to install, in addition to slowing down image acquisition ~10–100 fold, the total light throughput of channel-by-channel imaging is extremely poor (<10%). In the work presented here, we switched between filters within ~60 ms using a

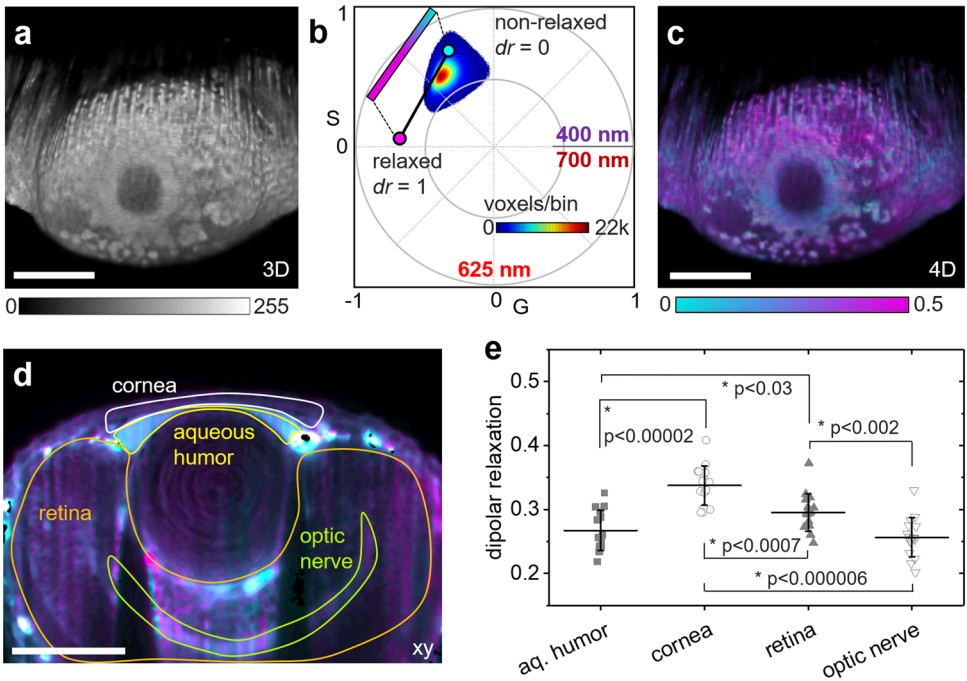

**Fig. 7 Cell-wide water dipolar relaxation in live zebrafish embryos. a–e** A live zebrafish embryo was incubated with the solvatochromic probe ACDAN before hyperspectral imaging with sine/cosine filters at 72 hpf. **a** MIP of 3D intensity image data obtained by two photon excitation with 800-nm light. **b** Spectral phasor plot obtained by fluorescence collection though sine/cosine filters. The dipolar relaxation index, $dr$, was quantified by calculating the voxel positions along the trajectory indicated in the phasor plot with the endpoints defined by the relaxed and unrelaxed state of ACDAN. **c** MIP of 4D spectral data color coded according to the voxel position along the phasor plot trajectory of ACDAN shown in **b**. **d** Single color-coded $z$ section of the 3D volume. **e** Quantification of the dipolar relaxation index, $dr$, obtained along the phasor plot trajectory in multiple regions of interest as indicated in **d**. Each point represents the average $dr$ in one $xy/yz$ slice. Error bars indicate the standard deviations. Stars (*) indicate significant ($p < 0.05$) differences between groups as determined by the Mann–Whitney U test, $N = 16$ $z$ sections each. Representative images of three different zebrafish embryos imaged on three different days. Scale bars, 100 μm.

motorized filter wheel, sufficient time resolution for the specimen under study. To avoid sample motion artefacts in extremely dynamic systems, sine/cosine filter imaging could be parallelized either by using multiple cameras or by splitting the emission on the same camera chip with commercially available multi-view hardware. We note that the current sine/cosine filters were designed to reach near-zero transmissions at phases of 270°/180° resulting in a lower accuracy of the spectral information obtained near those minima. This could be partially avoided by a filter redesign with 5–10% minimum transmission, the resulting shift in the spectral phasor coordinates could be corrected with an offset obtained from a calibration measurement. Theoretically, as opposed to resolving spectral information in discrete channels, the continuum provided by the sine/cosine filters provides the maximum possible spectral resolution of the phasor method, as shown in Supplementary Fig. S11. While we chose light sheet microscopy to implement the sine/cosine filter approach, super-resolution nanoscopy and tissue spectroscopy are other prime examples where hyperspectral imaging is highly desirable and the sine/cosine method can considerably enhance imaging capabilities. Finally, other filter design ranges could enable hyperspectral imaging in the ultraviolet and infrared with potential usage extending far beyond biological fluorescence including medical, industrial, and environmental applications.

## Methods

**Light sheet microscopy setup for hyperspectral imaging.** The light sheet microscope was based on our sideSPIM setup as previously described[21]. Briefly, a set of visible, fiber-coupled continuous wave lasers (405 nm, 488 nm, 561 nm, 638 nm, A461 Laser Launcher, ISS, Champaign, IL, USA) were combined with the free space near-infrared beam of a femtosecond pulsed Ti:Sa laser (Chameleon Ultra, Coherent, Santa Clara, CA, USA) via a long pass dichroic mirror (T670

LPXR, Chroma, Bellows Falls, VT, USA). The excitation beam was reflected with a scanning mirror assembly (A402, ISS) and relayed towards the excitation objective lens, either a 10x NA 0.3 (CFI Plan Fluorite, Nikon, Melville, NY, USA) or a 4x NA 0.1 (Plan Achromat, Olympus, Waltham, MA, USA), via a 50-mm focal length scan lens (#49-356, Edmund Optics Inc., Barrington, NJ, USA) and a 180-mm focal length tube lens (TTL180-A, Thorlabs, Newton, NJ, USA). The laser intensity of the NIR beam was adjusted by an acousto optic modulator (AA Opto-Electronic, New York, NY, USA) placed immediately after the laser output. The light sheet was injected into the sample from the side through the side window of the custom designed multi well chamber held in place with a magnetic sample holder inset. A motorized xy stage (MS-2000, ASI, Eugene, OR, USA) with a $z$ piezo top plate (PZ-2000, ASI) was used for sample positioning and $z$ stack acquisition. For detection of the fluorescence emission, either a 60x NA 1.0 (LUMPLFLN 60x/W, Olympus), a 40x NA 0.8 (LUMPLFLN 40x/W, Olympus), or a 20x NA 0.5 (LUMPLFLN 20x/W, Olympus) objective lens were used together with an sCMOS camera (Edge 4.2, PCO, Romulus, MI, USA) mounted to the inverted microscope frame (IX71, Olympus) which also contained a six position motorized fast filter wheel (FW103H, Thorlabs). The filter wheel was populated with sine, cosine, 460/80 nm, 540/50 nm, and 641/75 nm filters, one position was left empty to allow collection of the total intensity. Sine/cosine interference filters (170926DD01, 170926DD02) were custom ordered from OptoSigma (Irvine, CA, USA) with the transmission designed to resemble a single sine/cosine period within a range of 400-700 nm. The filter design specifications were 0° ± 5° angle of incidence, 25.4 mm diameter unmounted on 2 mm thick fused silica substrates with relative to theory manufacturing tolerances of ±5% relative to the transmission scale and ±0.5% relative to the wavelength scale, surface quality 60/40, clear aperture >90%, TWE: lambda/4 P-V at 632.8 nm. In addition, a 400–700 nm bandpass filter (170926DD03, OptoSigma) was added to the emission path in a fixed position to block any NIR excitation light and any light outside the design range of the sine/cosine filters. The filter spectra were verified by absorption spectroscopy measurements.

**Light sheet hyperspectral imaging.** In each plane, three images were acquired through the sine, the cosine, and without filter, respectively. The purpose of the images acquired without filter was to capture the total fluorescence intensity to normalize the spectral phasor data and to provide an unbiased intensity image. Fast, automated switching between filters (~60 ms) was achieved with a motorized filter wheel (FW103H, Thorlabs). For all images, a fixed bandpass filter in the

emission path ensured that light collection was limited to the design range of the sine/cosine filters of 400–700 nm. For single photon excitation of beads with 488-nm light, a 500 nm long pass filter (FF01-500/LP-25, Semrock, Rochester, NY, USA) was added to the emission path to block scattered laser light. For the solution measurements, we used the 10× NA 0.3 lens in excitation and the 60× NA 1.0 lens in detection (106 nm pixel size at the sample). Laser powers at the excitation lens were 0.5–2 mW for single photon excitation at 405 nm, 488 nm, 561 nm, and 638-nm. For live cell experiments, we used the 4× NA 0.1 lens in excitation and the 40× NA 0.8 lens in detection (159 nm pixel size at the sample) and excited with 870-nm light (20–100 mW at the objective). For the zebrafish and tissue measurements, we used 4× NA 0.1 in excitation and 20× NA 0.5 in detection (318 nm pixel size at the sample), laser powers at the excitation lens were 10–200 mW for two-photon excitation at 740–920 nm. Camera exposure time was 50–200 ms and filter switching time was ~60 ms resulting in hyperspectral imaging at 1–3 frames/second. Comparisons with conventional laser scanning microscopy (Fig. 1) were carried out on a commercial Zeiss LSM 880 (Jena, Germany) setup equipped with a Zeiss 32-channel spectral detector (410–695 nm); a pulsed NIR laser (MaiTai, Spectra-Physics, Santa Clara, CA, USA) was used for two-photon excitation with a 40× NA 1.2 water immersion lens.

**Solution samples**. Dye solutions of 9-10 Diphenyl-anthracene, Coumarin 1, ANS, Coumarin 6, Fluorescein, Rhodamine 110, Rhodamine 6 G, 5-TAMRA, Atto590, Alexa 594, NileRed, and Mitotracker Deep Red (Thermo Fisher Scientific, Waltham, MA, USA) were prepared in aqueous or ethanol buffers (Coumarin) at concentrations of 100 nM–1 μM. 9-10 Diphenyl-anthracene, Coumarin 1, ANS, and Coumarin 6 were excited with 405-nm light. Fluorescein, Rhodamine 110, and Rhodamine 6 G were excited with 488-nm light. 5-TAMRA, Atto590, and NileRed were excited with 561-nm light. Mitotracker Deep Red was excited with 638-nm light. Besides the 400–700 nm bandpass, no additional filters to block excitation light were needed for the non-scattering solution measurements because excitation and detection paths are intrinsically decoupled in light sheet microscopy. Each cluster on the phasor plot represents a 50-frame average (256 ×256 pixels, 50 ms camera exposure). In addition, aqueous solutions of NADH, FAD, and ACDAN were prepared and excited with 740-nm (NADH), 780-nm (FAD), and 800-nm light (ACDAN). Aqueous solutions of ACDAN (A168445, Toronto Research Chemicals, Canada) at 30 μM were used to pinpoint the pure species for the dipolar relaxation experiments in the zebrafish. Concurrently, 2 mM aqueous solutions of NADH and FAD (Sigma-Aldrich) were used to pinpoint the pure species for the redox ratio experiments in the mouse tissues.

**3D cell culture**. NIH 3T3 cells were plated on top of a ~1 mm thick layer of collagen type I, rat tail, (BD Bioscience, San Jose, CA, USA) in a custom-made chamber for the sideSPIM microscope. Briefly, on ice, the pH of collagen (9.73 mg/mL) was adjusted to pH 6.5–7.5 by slowly adding drops of a 1 M NaOH solution prepared in sterile 10X phosphate buffered saline (PBS). Collagen was diluted 1:4 with cell culture medium (DMEM GlutaMAX supplemented with 10% FBS and Pen-Strep) to 2.4 mg/mL, added to the imaging wells, and allowed to gel by 30 min incubation at 37 °C. Cultured cells were brought into solution by trypsinization, centrifuged, resuspended in cell culture medium, and added to the imaging wells. Cells were allowed to adhere to the collagen layers by overnight incubation at 37 °C and 5% CO₂. Cell organelles were fluorescently labeled by incubating with NucBlue (R37605, Thermo Fisher Scientific), BODIPY FL (D3521, Thermo Fisher Scientific), MitoTracker Orange (M7510, Thermo Fisher Scientific), and LysoTracker Red (L7528, Thermo Fisher Scientific) for 30 min. Live cell staining solutions were prepared according to the manufacturer instructions. Relative amounts added to the cells were adjusted to yield similar fluorescence intensities for all stains at 870-nm two-photon excitation. Cells were washed 3 times with preheated cell culture medium before subjecting the sample to sideSPIM hyperspectral imaging. The light sheet was generated with the 4x NA 0.1 objective lens while fluorescence was detected with the 40x NA 0.8 objective lens. Z-sections were acquired at 1 μm intervals, xy pixel size at the sample was 159 nm, camera exposure times were 200 ms.

**Zebrafish samples**. Zebrafish lines were maintained and bred at 26.5 °C. Embryos were raised at 28.5 °C and staged in hours post fertilization (hpf). Embryos were treated with 0.003% phenylthiourea (PTU, Sigma-Aldrich) at 8 hpf to delay pigmentation, and were anaesthetized by 0.04% MS-222 (Sigma) prior to live imaging or fixation. For fixed embryo experiments, 72 hpf SoFa zebrafish embryos (kindly provided by William Harris, University of Cambridge, UK) were fixed overnight at 4 °C with paraformaldehyde 4% in PBS, and washed 3 times in PBS with 0.5% Tween-20 (PBS-T). The samples were then incubated with SytoxOrange (S11368, Thermo Fisher) for 1 h at room temperature and washed 3 times with PBS-T before mounting in 1% low melting point agarose (T9284, Sigma–Aldrich) followed by light sheet imaging. For live embryo experiments, 60 hpf wildtype zebrafish (*Danio rerio*) were incubated with embryonic media (EM) containing 50 μM ACDAN for 12 h at 28°C. Embryos were removed from the labeling solution, anesthetized in EM with 0.0165% (w/v) tricaine mesylate (A5040, Sigma-Aldrich), and mounted in 1% low melt agarose (T9284, Sigma–Aldrich) in a custom-made chamber for the sideSPIM microscope. The zebrafish embryos were mounted parallel to the light sheet and perpendicular to the imaging plane and imaged 72 h post fertilization

(hpf). Fluorescent protein and SYTOX emissions were excited at 920 nm, ACDAN fluorescence was excited at 800 nm *via* two-photon excitation. The light sheet was generated with the 4x NA 0.1 objective lens while fluorescence was detected with the 20x NA 0.5 objective lens. Z-sections were acquired at 1 μm (fixed samples) and 1.5 μm intervals (live samples), xy pixel size at the sample was 318 nm, camera exposure times were 200 ms.

**Mouse tissue samples**. Tissues from male C57BL/6 N mice, obtained from the Knockout Mouse Project (KOMP) repository, were used. Mice were aged 9 weeks. Mouse colons (*caecum* to *rectum*) were removed, flushed, and linearized. The tissue was dissociated at 4 °C for 1 h in a solution of 2 mM EDTA and 10 μM Rhoassociated protein kinase (ROCK) inhibitor in PBS at slow rotation. Ten hard firm shakes then loosened the crypts and the suspension was filtered using 100-μm filters together with centrifugation at 500 × g for 5 min at 4 °C. The pelleted crypts were resuspended in Cultrex Matrix (Cultrex PathClear BME Type 2, Sigma–Aldrich) with organoid base media and ROCK inhibitor. Crypts were then plated in custom sideSPIM imaging chambers together with media composed of 50% organoid base media and 50% phenol-free DMEM, and imaged within hours. NADH and FAD fluorescence were excited with 740-nm light via two-photon excitation. The light sheet was generated with the 4x NA 0.1 objective lens while fluorescence was detected with the 20x NA 0.5 objective lens. Z-sections were acquired at 1 μm intervals, xy pixel size at the sample was 318 nm, camera exposure times were 200 ms. In the xy plane, pixels were binned 2×2 to increase signal-to-noise in the analysis.

**Data analysis**. The microscope hardware was controlled and image data was acquired with Micro-Manager 1.4[35]. Image data was analyzed and visualized with SimFCS 4 software available at http://www.globalssoftware.com/ (Globals for Images, Champaign, IL, USA) as well as scripts written in Matlab R2019a (MathWorks, Natick, MA, USA). 3D volumes were rendered with Fiji ImageJ 1.52p using the 3D Viewer Plugin. Data was graphed with OriginPro 2017 (OriginLab, Northampton, MA, USA).

**Statistics and reproducibility**. To normalize the spectral phasor coordinates, $G(x, y, z)$ and $S(x, y, z)$, transform the transmission ranges (0…1) to the ranges of the sine/cosine functions (−1…1), and to account for the non-zero camera offset, the following two equations (Eqs. 1,2) were used:

$$G(x, y, z) = 2\left(\frac{I(x, y, z)_{\cos} - b}{I(x, y, z)_{\text{total}} - b} - 0.5\right), \quad (1)$$

$$S(x, y, z) = 2\left(\frac{I(x, y, z)_{\sin} - b}{I(x, y, z)_{\text{total}} - b} - 0.5\right), \quad (2)$$

where $I(x, y, z)_{\sin}$ was the fluorescence intensity recorded through the sine filter, $I(x, y, z)_{\cos}$ the fluorescence intensity recorded through the cosine filter, $I(x, y, z)_{\text{total}}$ the total fluorescence intensity recorded without filter, and $b$ the camera offset. To remove salt and pepper noise introduced by hot and cold camera pixels, a 3 pixel median filter was applied to the image data. As the $G(x, y, z)$ and $S(x, y, z)$ values assumed random numbers in regions where no fluorescence was detected, sample-dependent intensity thresholds of 25-100 a.u. were applied to remove this unspecific background from the spectral phasor data. A random jitter in an interval of −0.02…0.02 was added to the G and S coordinates of each voxel to avoid binning artefacts in the resulting 2D histograms. For weak autofluorescence mouse tissue signals, the jitter was increased to −0.08…0.08 to avoid binning artefacts. Jitter was not used for any quantitative analyses such as optical redox ratio and dipolar relaxation calculations. After analysis, image data (16 bit) was transformed to 8 bit color maps as indicated in each figure. To aid visualization, the upper intensity cutoff was set at up to the brightest 5% of voxels. No such cutoff was used for any quantification. Linear color maps were used as indicated. Examples of raw camera image data are shown in the Supporting Information and are available as TIFF images upon request.

**Ethical approval**. All mouse and zebrafish work were performed in accordance with NIH guidelines and was approved by the Institutional Animal Care and Use Committee (IACUC) of the University of California, Irvine, review board approval number AUP-17-053. All experiments were carried out in accordance with relevant guidelines and regulations.

**Reporting summary**. Further information on research design is available in the Nature Research Reporting Summary linked to this article.

## Data availability
The raw data for charts and graphs in the main figures are supplied as supplementary data files. Additional information can be obtained from the corresponding author upon reasonable request.

## Code availability

The spectral phasor analysis is part of SimFCS software that is freely available at https://www.lfd.uci.edu/globals/ and http://www.globalssoftware.com/.

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

## Acknowledgements

We thank Irene Vorontsova and Danny Dranow from the lab of Thomas F. Schilling at UC Irvine for providing zebrafish embryos and Amber Habowski from the lab of Marian L. Waterman at UC Irvine for providing mouse tissues. We also thank Paola Lepanto and Gisell Gonzalez from the Zebrafish Facility at the Institut Pasteur Montevideo for providing SoFa embryos. This work was supported by NIH/NIGMS grants P41 GM103540 and R21 GM135493. L.M. was supported by the Agencia Nacional de Investigación e Innovación (ANII) by grant FCE_3_2018_1_149047 and the and Chan Zuckerberg Initiative.

## Author contributions

P.N.H. built the instrument and acquired data. P.N.H., L.M., R.C. and A.K. prepared samples and analyzed data. E.G. and P.N.H. wrote software for data analysis. P.N.H. wrote the manuscript. E.G. supervised the project.

## Competing interests

P.N.H. and L.M. have filed a patent for the sideSPIM technology under US Provisional Patent Application No. 62/456,298, WO2018148309A1. E.G. has filed the sine/cosine filter method under US Provisional Patent Application No. 62/853,237. The remaining authors declare no competing interests.
