## [Peer Review File · Communications Biology]

Reviewers' Comments:

Reviewer #1:

Remarks to the Author:

The submitted draft, describes a novel spectral imaging approach by directly capturing the spectral phasor data optically, using filters with sine, cosine transmissions. The proposed method is based on the previous work of the authors on spectral phasor approach where the multi-channel spectrum is cosine and sine transformed to build up the phasor plot. This approach has the benefit of skipping the multichannel spectral acquisition and capturing the phasor data directly.

The manuscript is well written and plenty of examples are provided.

- I have some reservation about using the term "snap-shot". This method needs 3 acquisitions: sine, cosine and blank transmission.

- My major concern is the lack of quantification in the manuscript. Authors need to show the sensitivity of the phasor filter approach over the regular RGB or 3 top hat filters. In other words, what happens, if I image the same field of view with RGB filters, top hat filters and phasor filters, within the same total exposure time, and create their corresponding phasor plots and compare their spectral resolving power? Authors really need to show the superiority of phasor filters over the conventional configuration by a more quantitative analysis.

Line 52:

- I would start with a general description to phasor approach itself, explaining that a complicated spectrum can be mapped on a 2D plot with pair of Fourier transforms. Then explain this transformation can be performed optically, without the need to capture the entire spectrum and by only 3 images.

Line 349: fix the units: "transmissions at phases of $270_{\circ}/180_{\circ}$ resulting in a lower accuracy"

Reviewer #3:

Remarks to the Author:

This manuscript describes the development of phasor-based hyperspectral snapshot microscopy for biomedical imaging, motivated by current hyperspectral imaging methods that are either slow or reducing light throughput. Some snapshot methods are fast, but are limited in spatial and/or spectral resolution, or computationally demanding, and devices are expensive to manufacture. Therefore, by simply replacing the standard emission filter with a sin/cosine filter, the method can be applied to any commercial or home-built imaging system including all camera-based and laser scanning microscopes without a specialized spectral detector. Importantly, their approach does not use computational unmixing.

I think the manuscript is well-written and the results are convincing. Therefore, I would recommend a minor revision of this manuscript to be accepted for publication. And I hope my comments listed below can help to improve the manuscript.

Main concerns:

1. The authors tend to say 3D top view or 3D intensity image in many figures. However, I think that is not precise enough. I would suggest saying it more specifically such as a MIP (maximum intensity projection), or a sum projection, or a single slice of a 3D image? For example, Fig. 3(c) is not 4D but rather a projection (MIP or SUM) or a single slice of a 3D image stack with the spectral information. And for example, fig. 3(d-h), is that a projection or single slice? The same concern applies to fig. 4, 5, 6, and 7.

2. Could you explain more how you determine where to cut into two halves in fig. 3b and suppl. fig. S5c? I think this is an important processing step through the manuscript.

Minor concerns:

1. Line 103-105, "In either case, image acquisition is slow as only a maximum of two of the dimensions (x,y, x,λ , or y,λ) required to form a full hyperspectral image (x,y, λ) can be acquired simultaneously." The sentence is strange to read, please improve it.

2. In fig. 1d, please clarify what are λ_1 , λ_2 , and n. Please also add coordinates to fig. 1f even it is obvious.

3. Please put figure titles in fig. 2C, such as "phase" and "modulation".
4. Line 165, 238, and 349, not sure if it's due to the conversion of the document but "°" is missing.
5. Line 166, I would suggest "zoomed-in of the white box in Fig. 3c".
6. Line 233 and line 694, I am still confused about what 3.5D exactly means? 3D image with not perfect separation of the spectrum? Could you describe it more specifically?
7. Figure 6, please add coordinates or specifically say that is a top view, .etc.
8. Materials and Methods, line 410, could you give more information on the 32 channel spectral detector, or is that also from Zeiss?
9. Materials and Methods, is zebrafish imaging also considered tissue imaging? If it is, please add in line 404, for the tissue and "zebrafish" measurements. If it is not, could you also provide the information for zebrafish imaging?
10. Material and Methods, line 408, I would suggest Comparisons with conventional laser scanning microscopy "(Fig. 1)" ..., just to be clearer for the readers.
11. Reference #13, line 561, the page should be 3503-3511, please check again.
12. I could not find any supplementary movies. Please verify if they are missing.

Reviewers' comments:

Reviewer #1 (Remarks to the Author):

The submitted draft, describes a novel spectral imaging approach by directly capturing the spectral phasor data optically, using filters with sine, cosine transmissions. The proposed method is based on the previous work of the authors on spectral phasor approach where the multi-channel spectrum is cosine and sine transformed to build up the phasor plot. This approach has the benefit of skipping the multichannel spectral acquisition and capturing the phasor data directly.

The manuscript is well written and plenty of examples are provided.

We thank reviewer 1 for taking the time to assess of our work and for providing feedback that is addressed in the revised manuscript.

- I have some reservation about using the term “snap-shot”. This method needs 3 acquisitions: sine, cosine and blank transmission.

We used the term snapshot to distinguish our method from other phasor-based hyperspectral imaging methods such as laser-scanning or tunable filter based approaches. We understand the reviewer’s perspective that more than one “snaps” are required if the method is implemented with a filter wheel, as described in this paper. However, we think the method still qualifies as snapshot approach. As we mention in the discussion, 3 cameras or an imager splitter could be used to consolidate all information into a single “snap”.

- My major concern is the lack of quantification in the manuscript. Authors need to show the sensitivity of the phasor filter approach over the regular RGB or 3 top hat filters. In other words, what happens, if I image the same field of view with RGB filters, top hat filters and phasor filters, within the same total exposure time, and create their corresponding phasor plots and compare their spectral resolving power? Authors really need to show the superiority of phasor filters over the conventional configuration by a more quantitative analysis.

The issue with the 3 top hat or “red”, “green”, “blue” filters is their large spectral width (~100 nm per channel for a typical color camera) with respect to the emission band of an average fluorophore (~40 nm). If the emission falls within one of those spectral channels, small spectral shifts/differences cannot be detected. This effect is shown in Supplementary Figs. S5 and S8. For example, the emissions of MitoTracker Orange (emission peak ~576 nm), and LysoTracker Red (emission peak ~593 nm) were within the same band of the “red” filter (641/75 nm) and thus could not be distinguished S8j. By measuring in a continuum (sin/cos), the spectral separation is independent of the emission peak position, and was separated in Fig. S8f,g. Another demonstration of the effect of using different channel numbers can be found in Fereidouni et al. 2012 (DOI: 10.1364/OE.20.012729).

To further address the comment of reviewer #1 and to systematically compare the spectral resolving power of phasor analysis with RGB and other imaging approaches, we added new Supplementary Fig. S11. For convenience, we copied the figure and figure text here:

Typical fluorescence emission spectrum of an organic dye was simulated shifting the peak emission in 10-nm steps in the detection range of 400-700 nm. The spectra were then split into multiple channels of varying number/width to represent typical imaging setups, namely “sin/cos” (300 channels of 1 nm – quasicontinuum representing our filters), “Multichannel PMT” (32 channels of 9 nm width – typical PMT-based hyperspectral detector), “6 Filter” (6 channels of 50 nm width – maximum capacity of a typical filter wheel), and “RGB” (3 channels of 100 nm width – typical color camera). On the spectral phasor plot, all possible positions for any emission spectrum can be found within a circle of radius=1 around the origin ($S=0, G=0$). The number of channels in the measurement defines the number of points this perfect circle is approximated with. Therefore, a continuum, such as provided by the sin/cos filters, provides the maximum possible spectral resolution. While not much difference can be observed for reducing to 32 channels, a significant portion of the phasor space becomes inaccessible with only 6 channels, and a stark reduction of the spectral resolution can be observed for a 3 channel “RGB” configuration. Basically, reducing the number of channels imposes limits on the number of linear combinations. The spectral resolving power remains high if the measured emission is positioned between two channels, but is strongly reduced as shifting to the center of one channel. This also illustrates the benefit of using hyperspectral detection in general polarization (GP) analysis over the conventional two-filter approach: While the sensitivity is high for positions of the emissions between the two filters, spectral shifts become indistinguishable at the end points. In conclusion, given the same imaging conditions (as it is the case with the simulations), the maximum spectral resolution of spectral phasor analysis is achieved with an infinite number of channels, i.e., a continuum as provided by the sin/cos filters. In practice, for most biological fluorescence imaging applications, 32 channels are sufficient. This is shown in Fig. 2 and Supplementary Fig. S2, where we measured the same samples with the sin/cos filters and a 32-channel detector (Zeiss LSM-880), effectively yielding the same S and G coordinates for all samples.

Line 52:

- I would start with a general description to phasor approach itself, explaining that a complicated spectrum can be mapped on a 2D plot with pair of Fourier transforms. Then explain this transformation can be performed optically, without the need to capture the entire spectrum and by only 3 images.

We changed the order of the introduction text to now first describe the phasor approach.

Line 349: fix the units: “transmissions at phases of $270_{\circ}/180_{\circ}$ resulting in a lower accuracy”

The units were correct in the original uploaded document and deleted by the conversion of the online submission system. We will work with the editor to fix this issue.

Reviewer #2 (Remarks to the Author):

This manuscript describes the development of phasor-based hyperspectral snapshot microscopy for biomedical imaging, motivated by current hyperspectral imaging methods that are either slow or reducing light throughput. Some snapshot methods are fast, but are limited in spatial and/or spectral resolution, or computationally demanding, and devices are expensive to manufacture. Therefore, by simply replacing the standard emission filter with a sin/cosine filter, the method can be applied to any commercial or home-built imaging system including all camera-based and laser scanning microscopes without a specialized spectral detector. Importantly, their approach does not use computational unmixing.

I think the manuscript is well-written and the results are convincing. Therefore, I would recommend a minor revision of this manuscript to be accepted for publication. And I hope my comments listed below can help to improve the manuscript.

Main concerns:

We thank reviewer 2 for carefully reading our manuscript and her/his recommendation for publication. Her/his comments are addressed below.

1. The authors tend to say 3D top view or 3D intensity image in many figures. However, I think that is not precise enough. I would suggest saying it more specifically such as a MIP (maximum intensity projection), or a sum projection, or a single slice of a 3D image? For example, Fig. 3(c) is not 4D but rather a projection (MIP or SUM) or a single slice of a 3D image stack with the spectral information. And for example, fig. 3(d-h), is that a projection or single slice? The same concern applies to fig. 4, 5, 6, and 7.

The reviewer is correct regarding the terminology and we now use the term maximum intensity projection (MIP) in the figures and the text. All image data shown are MIPs of 3D stacks or sub-volumes of 3D stacks. Regarding the terms 3D (x,y,z), 4D (x,y,z,lambda) and 5D

(x, y, z, λ, t), we used them as previously defined in Cutrale et al. Nat. Methods 14, 149–152 (2017).

2. Could you explain more how you determine where to cut into two halves in fig. 3b and suppl. fig. S5c? I think this is an important processing step through the manuscript.

We divided the phasor plot data into two halves, one including GFP/SYTOX the other one including CFP/RFP with the division running through the midpoints between GFP/CFP and SYTOX/RFP. We now specify this in the revised text.

Minor concerns:

1. Line 103-105, “In either case, image acquisition is slow as only a maximum of two of the dimensions (x, y, x, λ , or y, λ) required to form a full hyperspectral image (x, y, λ) can be acquired simultaneously.” The sentence is strange to read, please improve it.

We changed this sentence to: As only a maximum of two (x, y, x, λ , or y, λ) of the three dimensions required to form a full hyperspectral image (x, y, λ) can be acquired simultaneously, these methods are inherently slow.

2. In fig. 1d, please clarify what are λ_1 , λ_2 , and n . Please also add coordinates to fig. 1f even it is obvious.

We changed $\lambda[2]$ and $\lambda[1]$ to $\lambda[\max]$ and $\lambda[\min]$ for clarification and added coordinates to Fig. 1f.

3. Please put figure titles in fig. 2C, such as “phase” and “modulation”.

We added “phase” and “modulation” to Fig. 2c.

4. Line 165, 238, and 349, not sure if it's due to the conversion of the document but “°” is missing.

The units were correct in the original uploaded document and deleted by the conversion of the online submission system. We will work with the editor to fix this issue.

5. Line 166, I would suggest “zoomed-in of the white box in Fig. 3c”.

We changed to: zoomed-in 3D MIPs (white box in Fig. 3c) are shown in Fig. 3d-g

6. Line 233 and line 694, I am still confused about what 3.5D exactly means? 3D image with not perfect separation of the spectrum? Could you describe it more specifically?

Yes, that is correct, we now describe this in the text: 3.5D - 3D in few spectral channels

7. Figure6, please add coordinates or specifically say that is a top view, .etc.

We now specify “side view” in the figure.

8. Materials and Methods, line 410, could you give more information on the 32 channel spectral detector, or is that also from Zeiss?

The detector is also from Zeiss, we now mention this in the text.

9. Materials and Methods, is zebrafish imaging also considered tissue imaging? If it is, please add in line 404, for the tissue and “zebrafish” measurements. If it is not, could you also provide the information for zebrafish imaging?

Yes, we now added “zebrafish” in the text.

10. Material and Methods, line 408, I would suggest Comparisons with conventional laser scanning microscopy “(Fig. 1)” ..., just to be clearer for the readers.

We added “Fig. 1” to the text.

11. Reference #13, line 561, the page should be 3503-3511, please check again.

We checked and corrected the page numbers of this citation.

12. I could not find any supplementary movies. Please verify if they are missing.

We will work with the editor to fix this issue and make the movies available.

REVIEWERS' COMMENTS:

Reviewer #1 (Remarks to the Author):

With extra figures on supplementary parts, I think authors has addressed the raised comments. I can recommend this paper for publication.